# The Important Role of Zinc in Neurological Diseases

**DOI:** 10.3390/biom13010028

**Published:** 2022-12-23

**Authors:** Zhe Li, Yang Liu, Ruixue Wei, V. Wee Yong, Mengzhou Xue

**Affiliations:** 1Department of Cerebrovascular Diseases, The Second Affiliated Hospital of Zhengzhou University, Zhengzhou 450001, China; 2Academy of Medical Science, Zhengzhou University, Zhengzhou 450001, China; 3Henan Medical Key Laboratory of Translational Cerebrovascular Diseases, Zhengzhou 450001, China; 4Department of Clinical Neurosciences, Hotchkiss Brain Institute, University of Calgary, Calgary, AB T2N 1N4, Canada

**Keywords:** zinc, brain, stroke, neurotoxicity, brain injury

## Abstract

Zinc is one of the most abundant metal ions in the central nervous system (CNS), where it plays a crucial role in both physiological and pathological brain functions. Zinc promotes antioxidant effects, neurogenesis, and immune system responses. From neonatal brain development to the preservation and control of adult brain function, zinc is a vital homeostatic component of the CNS. Molecularly, zinc regulates gene expression with transcription factors and activates dozens of enzymes involved in neuronal metabolism. During development and in adulthood, zinc acts as a regulator of synaptic activity and neuronal plasticity at the cellular level. There are several neurological diseases that may be affected by changes in zinc status, and these include stroke, neurodegenerative diseases, traumatic brain injuries, and depression. Accordingly, zinc deficiency may result in declines in cognition and learning and an increase in oxidative stress, while zinc accumulation may lead to neurotoxicity and neuronal cell death. In this review, we explore the mechanisms of brain zinc balance, the role of zinc in neurological diseases, and strategies affecting zinc for the prevention and treatment of these diseases.

## 1. Introduction

Zinc is a trace element essential for human survival, and its deficiency has been linked to various adverse effects, such as growth retardation, impaired functioning of the immune system, and cognitive dysfunction [1,2]. Zinc is associated with the synthesis and activity of numerous proteins and enzymes, including matrix metalloproteinases (MMPs), deoxyribonucleic acid (DNA) and ribonucleic acid (RNA) polymerases, and insulin [3]. Zinc insufficiency is one of the world’s major public health issues since it is linked to several diseases [4,5,6,7]. Maintaining zinc homeostasis is crucial for normal brain function, and zinc deficiency or overload may contribute to brain injury and exacerbate neurological conditions [8].

Zinc plays a role in both physiological and pathological processes in CNS [9,10]. A major site of storage of zinc is the telencephalon, especially in the cortical areas, hippocampus, and amygdala [11]. Accordingly, zinc has far-reaching effects on cognition, emotional stability, and memory [12]. Therefore, maintaining zinc homeostasis is essential for brain health, and it appears relevant to investigate its potential contributions to many neurological diseases. In this review, we provide a quick overview of zinc regulation and then focus on the roles that zinc plays in diverse brain disorders, as well as recent advances in zinc-related therapies.

## 2. The Homeostasis of Zinc in Physiological Conditions

The adult human brain contains approximately 0.04 g of zinc, accounting for 1.5% of the total body zinc [13,14]. The majority of zinc in the human body is located in zinc-containing proteins. A bioinformatics analysis in humans found that over 2800 proteins are potentially zinc-binding, constituting about 10% of all proteins in the human proteome; these zinc-binding proteins fulfill signaling, catalytic and structural functions [15]. Most zinc-binding proteins feature zinc-finger motifs, which occur when a zinc ion binds four invariant cysteines and/or histidine residues to form a stable structure, regulating protein-DNA [16], protein-RNA, and protein-protein interactions [17]. The physiological roles of zinc in cells and organs are summarized in Figure 1.

Zinc ions (Zn^2+^), which can be detected histochemically and eliminated by chelating agents, are abundant in all organs, tissues, body fluids, and secretions in addition to being tightly linked to proteins [1,18]. Zinc is highly concentrated (>95%) in fat-free body masses, mainly in intracellular compartments [19]. Under physiological conditions, zinc ions and zinc-binding proteins comprise the zinc pool in human tissues and maintain zinc homeostasis [20].

Zinc is a structural or functional component of many proteins [21]. It is an essential cofactor in protein structure stabilization and enzyme catalyses, such as DNA synthesis, brain development, and neurotransmission [22,23]. Because of its multifaceted role in biological processes, changes in zinc concentrations from normal levels can contribute to a variety of devastating diseases. For instance, zinc deficiency is associated with decreased immunological responses and growth retardation, in addition to neurological disorders [24,25,26].

## 3. Distribution of Zinc in the Brain

At 2–3 g in total, zinc is an abundant transition metal found in high concentrations in the mammalian brain and is distributed unequally throughout different organs and tissues [26]. In adults, 60% of zinc is found in skeletal muscles; 30% in bones; 5% in the liver and skin; 1.5% in the brain, kidney, and heart; and less than 2% in other tissues [27]. Only a small fraction of zinc circulates in the blood, approximately 80% of which is loosely bound to albumin, and 20% is tightly bound to α2-macroglobulin [28]. Zinc is crucial for the development and physiology of the mammalian brain [29]. The amount of zinc in brain tissue is second only to iron in terms of trace metal concentration [20]. In this regard, while the iron content in the normal brain is around 0.04 mg/g with a concentration of ~ 720 μM, that of zinc is about 10 μg/g with a concentration of 150 μM [30,31]. Timm-Danscher and Nissl stainings were used to analyze the distribution of zinc in the rat brain and revealed that zinc was highly deposited in neuron-rich areas, primarily the hippocampus and cerebral cortex [32,33]. 

The brain stores zinc in three different forms: vesicular zinc, protein-bound zinc, and zinc ions (Zn^2+^) [32]. In the CNS, around 85% of zinc is tightly linked to proteins that perform both functional and structural purposes, such as zinc-related metalloenzymes, MMPs, and zinc transcription factors [34,35,36]. Vesicular zinc is mostly present in the synaptic vesicles at the axon terminals of glutamatergic neurons, where it is released in a calcium- and the impulse-dependent way [37]. The concentration of Zn^2+^ in the brain is roughly ten times higher than that in serum, and it is required for neural regulation, synaptic plasticity, learning, and memory [34]. 

Given that zinc is unable to diffuse across the cell membrane, specialized transporters or membrane channels are required for its entry into cells [38]. Zinc transporters mainly include metallothioneins (MTs), Zrt-Irt-like proteins (ZIPs), and the zinc transporter family (ZnTs). Zinc enters neurons through three main pathways: α-amino-3-hydroxyl-5-methyl-4-isoxazolepropionate-acid /kainite (Ca^2+^-A/K) AMPA/KA channels [39], voltage-dependent channels, and N-methyl-D-aspartate (NMDA) receptor-gated channels [40]. In particular, AMPA/KA gated channels can pass more zinc in neurons [41,42,43].

### 3.1. Role of Zinc in Neurogenesis

Neurogenesis is particularly robust in embryonic and neonatal periods. Zinc plays critical roles in neuronal proliferation and differentiation, neuronal migration, and axonal growth during neurodevelopment [30]. It is necessary for the synthesis of a wide range of proteins, hormones, and growth factors [44]. During embryogenesis, stem cells and neuronal progenitor cells proliferate and differentiate in the neural tube, one of the first brain structures. Zinc has been recognized to be essential for DNA polymerase and other crucial enzymes in the developing brain [33,45]. As a result, zinc shortage during pregnancy leads to abnormal hippocampal proliferation and neuronal differentiation in the fetus, which appears to impact future memory and learning processes and even neurological abnormalities [46,47].

However, neurogenesis is not limited to the developmental phase, and zinc is required for the proliferation, differentiation, and death of neurons throughout the human brain’s life cycle [44]. Wistar females, during pregnancy and lactation, received a drinking water solution of ZnSO4 at an estimated dose of 16 mg/kg. Behavioral tests showed enhanced spatial memory and increased blood zinc levels in the pups [48]. The US recommended dietary allowance for zinc for women during pregnancy is 11.5 ± 1.75 mg/d [49]. Zinc deficiency (<56 μg/dL) during pregnancy and breastfeeding can impede the growth of newborns, increasing the risk of dwarfism, impaired learning memory function, and delayed mental development, whereas zinc supplementation can improve the development of premature newborns [50,51].

### 3.2. Role of Zinc in Promoting Redox Homeostasis

Several age-related chronic illnesses, including atherosclerosis, cancer, and dementia, display oxidative stress as a contributing component. Reactive oxygen species (ROS) are chemically reactive chemical entities containing oxygen, such as O_2_·-, H_2_O_2_, and OH, which are constantly produced in vivo under aerobic conditions [52,53]. Zinc is a redox-inert metal that participates in redox-regulated signaling. It acts as an antioxidant by accelerating the action of copper/zinc superoxide dismutase [54], maintaining membrane structure, and promoting the synthesis of metallothionein (MT), a metal-binding protein [55]. Zinc helps to maintain cellular redox equilibrium via a variety of processes, including zinc’s dynamic interaction with sulfur in protein cysteine clusters [56], control of oxidant generation and metal-induced oxidative damage [57], and controlling redox signaling by altering enzyme activity, binding interactions, and molecular chaperone activity [58].

Given the complexity of the pathways and events that disruption of zinc homeostasis might influence, the negative effects of these changes are not to be underestimated. The events involved in the regulation of cellular oxidative/antioxidant homeostasis by zinc are diverse and interrelated, including (i) regulation of oxidant production and metal-induced oxidative damage; (ii) regulation of glutathione (GSH) metabolism and overall thiol redox status by zinc; and (iii) direct or indirect regulation of redox signaling [52]. Thus, the dysregulation of zinc homeostasis may have significant adverse effects. Deficits in zinc are associated with oxidative stress, impaired GSH metabolism, tubulin oxidation, and disruptions in redox-sensitive signaling in the developing nervous system [59]. These alterations may lead to altered organ cellularity, organization, and connectivity, thereby increasing the risk of diseases later in life [60,61].

### 3.3. Role of Zinc on Immunity in the CNS

Patients with zinc deficiency show increased susceptibility to various pathogens [62]. Regarding the human immune system, the innate and adaptive immune systems are both regulated by zinc-finger-bearing transcription factors. Hence, a direct, as well as an indirect role of zinc in altering intracellular signaling, can be anticipated [63]. The role of zinc in immune function is also connected to zinc transporter proteins. Overall, even a minor zinc shortage may elevate pro-inflammatory cytokines, including tumor necrosis factor-alpha (TNF-α), interleukin 1 beta (IL-1β), and IL-6 [64,65,66]. 

Microglia are an innate immune cell type found in the CNS. In response to insults, such as infection or other detrimental molecules, microglia are activated and secrete MMPs, ROS, and other pro-inflammatory molecules [67,68]. In models of ischemic stroke and neurodegenerative diseases, zinc chelator TPEN has been shown to prevent neuronal loss by inhibiting microglial activation, which is triggered when neurons release zinc [69]. PARP-1, which can chelate zinc, has been shown to minimize neuronal death in rats with ischemic stroke [70] and other neurodegenerative diseases [71].

## 4. The Role of Zinc in Stroke

### 4.1. Ischemic Stroke

As the primary cause of disability, stroke ranks just behind heart disease as the second biggest cause of mortality globally [67]. Ischemic stroke accounts for approximately 80% of all strokes and may be caused by a combination of events such as cardiogenic embolism, obstruction of tiny blood arteries in the brain, and atherosclerosis influencing cerebral circulation [72]. Ischemic stroke is a thrombo-inflammatory disease in which platelets and immune cells accumulate at sites of ischemic vascular damage, disrupting the permeability of the BBB, triggering other processes such as neuroinflammation, microglia activation, and excitotoxicity; these collectively contribute to neuronal death [67,73].

Zinc may provide anti-atherogenic properties by preventing metabolic and physiological dysregulation of the vascular endothelium due to its antioxidant and membrane-stabilizing characteristics [74,75]. The supplement, along with zinc in zinc-deficient endothelial cells, induces a partial repair of the endothelial cell barrier, but supplementation with calcium and magnesium does not achieve the same effect [76]. As a consequence, zinc appears to be required for endothelial integrity, and a deficit might compromise endothelial barrier function [77]. 

During ischemic stroke, zinc is released from synaptic vesicles of glutamatergic neurons, and the abnormal accumulation of zinc stimulates the activation of microglia [78] and ultimately influences cell survival and function in part through mitochondrial dysfunction [79]. In ischemic stroke, zinc release can directly produce ROS or activate NADPH oxidase to produce ROS, leading to brain injury [17] (Figure 2). Zinc is considered an independent risk factor for ischemic stroke [75,80], and it has been observed to accumulate in the synaptic gap of neurons in animal models of ischemic stroke, speeding up the development of cerebral infarction [81]. As a result, inhibiting zinc over-release may prevent brain injury after ischemic stroke.

In vitro and animal experiments support a causal relationship between zinc dysregulation and neuronal damage after ischemic stroke. The injection of the zinc chelator ZnEDTA 14 days after middle cerebral artery occlusion (MCAO) in adult male rats led to a significant decrease in infarct volume and neuronal damage and improvement in neurological function [82]. In MCAO rats, zinc-induced CDK5-Tyr15 phosphorylation activates CDK5 in the hippocampus, which exacerbates neuronal death in ischemic stroke [83]. In an experimental model of ischemic stroke, normobaric hyperoxia treatment reduces zinc accumulation in penumbral tissues, thus reducing ischemic injury [84]. 

Some noteworthy findings about zinc levels have been reported from clinical investigations of patients diagnosed with an ischemic stroke. Serum zinc concentrations were considerably lower in ischemic stroke patients compared to age- and sex-matched healthy controls [85]. Calcium, copper, and iron levels did not change significantly between patients with ischemic stroke and healthy controls; the researchers postulated that low blood zinc concentration is associated with an increased risk of ischemic stroke based on their findings [86]. In a case-cohort study, blood zinc concentrations were inversely linked with the incidence of ischemic stroke, especially in women. This might be related to the function of zinc in the metabolism of sex hormones and the reproductive cycle, but the mechanism underlying the possible interaction is still unclear and warrants further investigation [87]. It appears that ischemic stroke could be a condition that would benefit from preventative zinc supplementation [86].

### 4.2. Intracerebral Hemorrhage (Hemorrhagic Stroke)

Intracerebral hemorrhage (ICH) is a non-traumatic hemorrhage in the brain parenchyma. ICH is particularly catastrophic, with a high mortality rate of up to 50% of the survivors. Over 70% are dependent on functioning aids a year after the event [88,89,90]. Despite clinical advances, including the continued evolution of minimally invasive surgical procedures to remove the blood clot, the prognosis of ICH remains poor [91,92,93]. A number of factors play a role in the pathogenesis of ICH, including hypertension, diabetes, lipid metabolism problems, and genetics, but also changes in the levels of essential trace elements and heavy metals [80,94,95,96]. 

Zinc is involved in coagulation’s intrinsic pathway, where it binds directly to the XII factor and increases its sensitivity to enzymatic activation [97]. Therefore, both animal models and patients with zinc deficiency have been shown to have prolonged bleeding times as a result of impaired coagulation cascades and fibrin formation [98]. Similarly, after platelets are activated at the site of injury, zinc binds to fibrinogen and fibrin, promoting the formation of a fibrin network [98,99]. Patients with ICH had lower zinc levels than normal controls (0.13 ± 0.02 vs. 3.17 ± 0.74 μg/dI); *p* < 0.001) [100]. It has been determined that plasma zinc levels and the risk of first-ever stroke in hypertensive patients are significantly correlated with the risk of ICH in a recent study [101]. Previously, a retrospective cohort study examined the relationship between hypozincemia and the severity of aneurysmal subarachnoid hemorrhage. The findings indicated that zinc deficiency (plasma zinc concentration < 10 μmol/L) is associated with a more severe course of the disease. As a result, the authors concluded that zinc deficiency independently contributes to an unfavorable outcome [102].

## 5. Zinc and Other Disorders of the CNS

### 5.1. Alzheimer’s Disease

Alzheimer’s disease (AD) is a neurodegenerative condition that affects over 37 million individuals throughout the globe and is triggered by a wide range of risk factors [103]. Neuropathological diagnosis of AD requires the presence of two primary indicators: the presence of amyloid-beta (Aβ) protein plaques outside neurons and the presence of deposits of neurofibrillary tangles (NFTs) within neurons, both of which lead to neuronal degeneration in the hippocampus, amygdala, and neocortex [104,105]. 

AD is a neurodegenerative condition characterized by a gradual but steady deterioration in cognitive abilities, such as cognitive decline, language impairment, mental and physical disorientation, and altered behavior [106]. Depression, anxiety, and agitation are just some of the non-memory-related behavioral symptoms it produces [107]. There was a twofold increase (25 μg/g) in zinc levels in AD brain tissue as compared with normal age-matched control subjects [108]. Brain zinc accumulation is a prominent feature of advanced AD. During the development of AD, there are unexpected changes in the expression levels of the zinc transporter, which includes six families of the zinc transporter (ZnT) and one family of the Zrt-/Irt-like protein (ZIP) transporter. There were changes in ZnT proteins (ZnT-1, ZnT-4, and ZnT-6) observed in AD patients [109]. ZnT1 and ZnT4 were expressed across the whole senile plaque, whereas ZnT3, ZnT5, and ZnT6 were localized to the plaque’s perimeter [110]. Genetic ablation of ZnT3 may reflect a phenotype of synaptic and memory deficiencies in AD since ZnT3 knockout animals have been shown to have problems in learning and memory [111].

Amyloid deposits of aggregated Aβ protein and neurofibrillary tangles (NFT) are pathological characteristics of AD. Zinc-binding sites may be found in both the amyloid precursor protein (APP) and Aβ [112]. Zinc transporters have been implicated in the development of AD senile plaques by several studies [106,109,113]. In senile plaques of the human Alzheimer’s disease brain and APP or APP/presenilin (PS) animal models, zinc transporters were discovered to be overexpressed and distributed differently; an extracellular elevation of zinc concentration may initiate the deposition of Aβ and lead to the formation of senile plaques [106,114]. Excess chronic zinc supplementation is also linked to hyperphosphorylated tau aggregation in AD pathogenesis at the molecular level [115].

The effect of excessive chronic zinc supplementation on tau pathology was assessed using a transgenic mouse model containing the human tau (P301L) gene [116]. In tauopathic mice, zinc supplementation aggravated circadian rhythm deficits, nesting behavior, and the Morris water maze deficits, suggesting zinc exacerbates tauopathic and biochemical deficits [116]. Tau hyperphosphorylation and other metabolic abnormalities have been attributed to zinc’s inhibition of proteins such as protein phosphatase 2A (PP2A), followed by activation of extracellular signal-regulated kinase (ERK), glycogen synthase kinase-3 (GSK-3) and other pathways [117]; these culminate in a characteristic reduction in microtubule stability in neurons and ultimately neurodegenerative morphological abnormalities [118].

Overall, zinc dyshomeostasis may play a crucial role in AD, making the discovery of an appropriate agent for zinc chelation a promising therapeutic target.

### 5.2. Traumatic Brain Injury

Traumatic brain injury (TBI) is a widespread public health and socioeconomic issue that affects about 70 million people worldwide annually and is a leading cause of disability and fatality [119]. TBI patients have a variety of behavioral abnormalities, including impairments in motor, sensory, attention, memory, and executive skills [120,121]. The major cause of TBI is mechanical damage to brain tissue induced by external forces (impact, nerve axon rupture, vascular injury, etc.), and the initial phases are associated with reduced cerebral blood flow, oxygen consumption, and metabolism [122]. Secondary injuries may cause brain edema, tissue atrophy and death, neuroinflammation, oxidative stress, mitochondrial dysfunction, and other harmful chain reactions [123]. The degree of damage is determined by the ultrastructure and mechanical injury to the brain and can occur from minutes to months [121].

The function of zinc in TBI has been the subject of numerous ideas throughout the last three decades of study. Following TBI, glutamate is released from presynaptic terminals, and intracellular zinc accumulates inappropriately in postsynaptic neurons, resulting in excitotoxicity and cell death [124]. As a result, researchers have started investigating whether zinc chelators could be useful in reducing the severity of neuronal damage caused by TBI. Previous research has demonstrated that as zinc chelators, clioquinol (CQ) [125] and N,N,N′,N′-tetrakis-(2-pyridylmethyl) ethylenediamine (TPEN) [126] may regulate zinc overload after TBI and have the potential to be therapeutic or preventative for TBI. 

During pathological conditions, excess zinc is rapidly released from presynaptic neuronal vesicles, crosses the postsynaptic membrane via channels or transporters (translocation), and causes neuronal damage and death. Recent studies have shifted the focus from this hypothesis to zinc chelation, where zinc is released and induces neuronal death when hippocampal neurons are overexcited [127]. According to preliminary in vitro investigations, zinc deficiency inhibits the proliferation of neural precursor cells and triggers death through p53 [128] and cysteine-mediated pathways [129]. Similar to this, dietary zinc deficiency (zinc contents in the zinc-deficient and zinc-adequate control diets were 0.85 ppm and 30 ppm, respectively) in rodents for 5 weeks inhibited the proliferation of neural precursor cells and hippocampus neurogenesis in animal studies [33,130]. Furthermore, as mentioned above, neurogenesis was considerably decreased in ZnT-3 deficient animals lacking presynaptic vesicle zinc [111]. As a consequence, research in recent years has focused on the possible neuroprotective effects of zinc supplementation. In a clinical trial, 68 patients with severe brain injury were randomly allocated to the zinc supplementation (12 mg elemental zinc) versus the routine zinc therapy group (2.5 mg elemental zinc), both as the sulfate salt in their total parenteral nutrition solution. One month after injury, the Glasgow Coma Scale (GCS) scores considerably improved, and the death rate was cut in half (26% vs. 12%) in the zinc supplementation group [131]. Similarly, animal studies have demonstrated that dietary zinc therapy may boost TBI resistance and enhance behavioral outcomes such as anxiety, sadness, spatial learning, and memory [132].

### 5.3. Epilepsy

Epilepsy is a common neurological disorder characterized by chronic brain seizures caused primarily by abnormal, excessive, or synchronized neuronal activity [133]. More than 60 million people worldwide suffer from epilepsy, which can be brought on by a variety of factors such as genetics, abnormal brain development, medications, and head trauma [134,135]. According to epilepsy research data, zinc’s neuroprotective and neurotoxic effects appear to be dose-dependent, and it can have both pro- and anticonvulsant effects [136,137]. Postnatal day 27 (P27) male Sprague-Dawley rats pretreatment with a zinc-deficient diet (2.7 mg/kg) for 4 weeks had long-term adverse effects of seizures. In contrast, zinc supplementation (246 mg/kg) for 4 weeks significantly mitigated the severity of pilocarpine-induced limbic seizures and prolonged the time it took for forelimb contracture to occur. High-zinc dietary treatment improved cognitive impairment and reduced regenerative sprouting of hippocampal mossy fibers [138,139]. 

In addition, zinc chelation reduced the frequency of EEG spikes and the duration of both behavioral seizures and electrical after-discharges. While this did not alter the severity of behavioral seizures, it does suggest that zinc may play a facilitative role in the onset of epilepsy [140]. Treatment with zinc with cyclo-(His-Pro) enhances vesicular zinc levels as well as hippocampus neurogenesis in rats [141]. According to a case-control study, children with intractable epilepsy had considerably lower blood zinc levels than healthy children. One possible explanation for this is that low zinc levels increase oxidative stress and the incidence of seizures since zinc is a component of glutathione peroxidase and superoxide dismutase [142]. 

Consequently, before considering zinc supplementation as a therapeutic method for the treatment of epilepsy, it is essential to investigate the anticonvulsant or antiepileptic effects of proteins related to zinc signaling.

### 5.4. Depression

Depression is a common chronic mental disorder that has risen to become the world’s second most common disorder, with symptoms primarily manifesting as changes in mood, sleep, cognition, and appetite [143,144]. Overall social functioning is impaired in patients with depression, and the vast majority of them do not receive an accurate and timely diagnosis, making depression one of the leading causes of suicide and disability [145,146]. Despite the lack of clarity surrounding the pathogenesis of depression, numerous studies have revealed that the majority of patients with major depressive disorder (MDD) have decreased serum zinc levels [147]. One possible mechanism involved in the changes in hippocampal functions, behavior, and pathophysiology observed in zinc deficiency maybe the activation of the hypothalamo-pituitary-adrenocortical (HPA) system [148].

Three decades ago, the association between zinc and depression was hypothesized for the first time [149]. In subsequent animal experiments, zinc deficiency was found to increase depression-like behavior in rodents during forced swimming and tail suspension tests [150,151,152]. Recent research using the olfactory bulbectomy model (OB) of depression in rats has revealed interesting results, whereby a zinc-deficient diet (3 mg Zn/kg) exacerbated depression. Importantly, treatment with escitalopram, venlafaxine 10 mg/kg, i.p., or combined escitalopram/venlafaxine (1 mg/kg, i.p.) with zinc (5 mg/kg) for 3 weeks reversed the behavioral changes of depression, indicating that zinc deficiency may lower the efficacy of anti-depressant drugs [153].

Human studies corroborated the above finding, showing an inverse relationship between serum zinc levels and the severity of depression. A meta-analysis revealed that zinc concentrations were approximately 1.85 µmol/L lower in depressed patients, and the severity of depression correlates with the relative zinc deficiency [154]. Furthermore, similar findings were observed in depressed patients such as postmenopausal women (72 ± 14 μg/dL) [155], college-aged women (79.6 ± 30.7 μg/dL) [156], and hemodialysis patients with end-stage renal disease (67.46 ± 29.7 μg/dL) [157]. Additionally, women with poor dietary or supplementary zinc consumption were more likely to suffer depressive symptoms, whereas men had no significant correlation [158].

## 6. Summary and Conclusions

Zinc plays a critical role in all life forms, in which zinc homeostasis is required for catalysis, structure, and regulatory processes. Zinc deficiency is commonly caused by poor dietary intake, while abnormal zinc accumulation is associated with the inhibition of zinc transporters or the activation of oxidative stress. Hence, zinc homeostasis and its regulation may affect a wide range of molecular functions and pathological outcomes, and different environmental and dietary habits and ecological conditions may influence zinc homeostasis.

Zinc has a profound effect on the brain throughout human life, from the development of the neonatal brain to neurological conditions. Zinc homeostasis in the CNS is regulated by a complex interplay of components, including the permeability of the BBB, the activity of zinc-binding proteins and ion channels, and the activity of various transporters (particularly ZIPs). Zinc is a signaling molecule in numerous metabolic pathways, the coordination of which occurs through the activity of zinc transporters. The surge in the development of a full suite of new fluorescent Zn^2+^ sensors, chelators, and genetic mouse models allows us to gain a greater understanding of neuronal signaling of Zn^2+^ in live primary neuron culture, brain slices, and live animals. In this review, we look at the evidence linking zinc metabolism to common neurological diseases such as stroke, Alzheimer’s disease, epilepsy, and TBI in animal and human studies.

Regulating zinc levels in the brain has emerged as a potential treatment target for neurological disorders. It is now clear that the development of new strategies for the treatment of CNS insults needs to take into account the role of zinc in neuronal function, damage, repair, apoptosis, and necrosis. Excessive zinc is released following brain injury in conditions including ischemic stroke and epilepsy, causing neurotoxic damage to neurons and glial cells. Given this, zinc chelators may be used as a preventative therapy for diseases exacerbated by excess zinc, although removing zinc from its dependent enzymes may be undesirable. Regardless of the approach, systemic zinc supplementation or central zinc chelation, future work that seeks to explore the possible efficacy of zinc modulation in stroke, TBI, and seizure disorders should employ clinically relevant models to permit not only the development of drug therapies but also the clarification of dietary recommendations after injury. 

Nevertheless, there is still room for improvement in clinical research on zinc chelators and zinc supplements. While nanoparticle delivery holds the best promise for restoring neural zinc imbalances, methods to therapeutically manipulate zinc levels in the brain remain elusive. These limitations are contributed by several factors, such as serum zinc measurements being non-indicative of brain zinc concentrations, the efficacy of supplemental zinc absorption and transport, the lack of therapeutic ways to manipulate zinc levels in the brain, and most importantly, nanoparticle compounds must be directed past the BBB in order to reach their intended targets.

It is important to take note that zinc’s functions in the central nervous system are quite diverse, but all of its actions are linked to homeostasis. Thus, in order to utilize zinc as a therapeutic agent, it is necessary to have a detailed understanding of the factors that influence zinc homeostasis as well as the ability to optimize the relevant concentrations. There is an ongoing research effort investigating the tight regulatory balance between bound zinc and labile zinc, which can influence many molecular functions and pathological outcomes. Even though significant progress has been achieved in deciphering the processes that underlie zinc’s metabolism and function, there is still a significant amount of information that has to be uncovered at the frontier of zinc neuroscience.

## Figures and Tables

**Figure 1 biomolecules-13-00028-f001:**
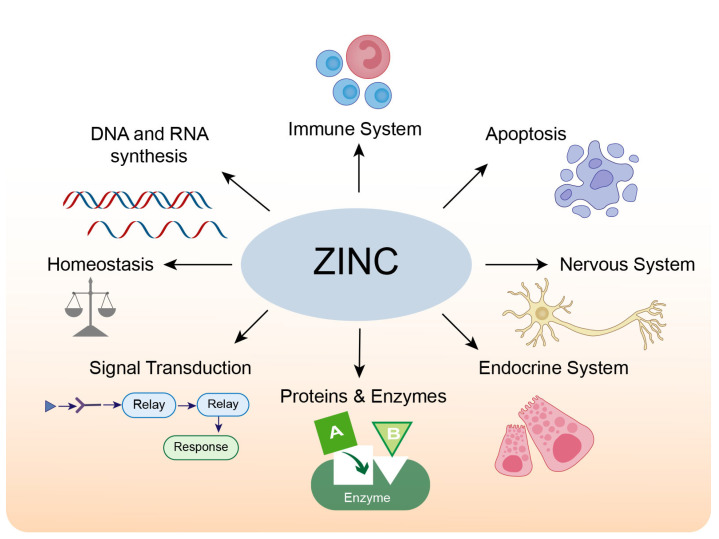
The physiological roles of zinc.

**Figure 2 biomolecules-13-00028-f002:**
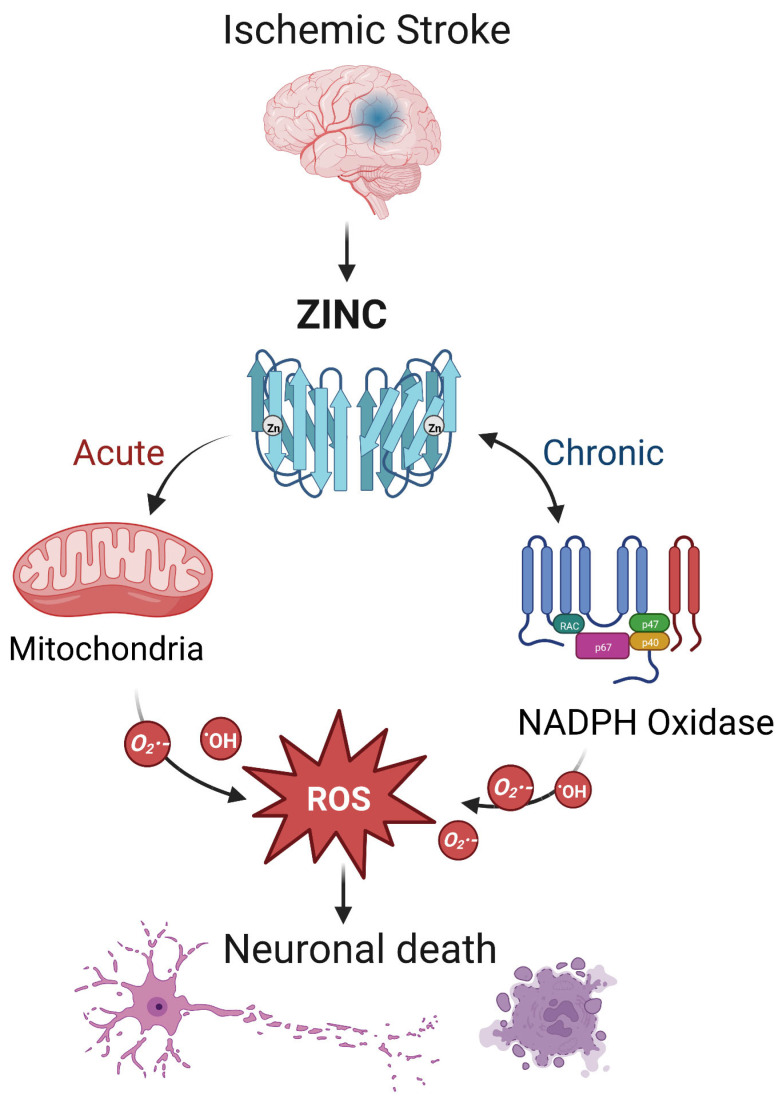
The interaction between zinc and ROS in ischemic stroke. In ischemic stroke, zinc is released from sources such as the synaptic vesicles of glutamatergic neurons. Acutely, zinc may elevate ROS through action on mitochondria. In the chronic phase, zinc accumulation can activate neuronal NADPH oxidases to generate additional oxidative species. The resultant ROS may produce neurotoxicity, further increasing zinc accumulation and ischemic brain injury. Note that while we show a central role for ROS, other mechanisms of zinc toxicity exist and are described in the text.

## Data Availability

The data in this study could be available from the corresponding author.

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
