# Peer review of "The Important Role of Zinc in Neurological Diseases"

_biomolecules, 2022, doi:10.3390/biom13010028_

Round 1
Reviewer 1 Report
The review by Li et al., ‘The emerging role of zinc in neurological diseases’ has tried to address the role of zinc in various neurological conditions and highlight how zinc can be targeted for therapeutic outcomes. Although the topic is relevant, but there are certain aspects which the authors have to be careful. The quality of the manuscript needs to be worked upon, since several section are redundant and sometimes contain misleading aspects.
1. Novelty aspects: Although a review is always great to read, the authors must take what they are offering in their manuscript! Reviewing the role of zinc in various physiological and neuropathological conditions has been done many times in the past (PMID: 22473811, 22072659, 22457776, 33255662, 7082716, 23882214, and many more) and also some in the present special issue "Zinc in Health and Disease Conditions". The authors did not put forward a solid motivation for writing this review, which in such a scenario becomes redundant!
2. Irrespective of what is mentioned above, the present manuscript has many disparities. Starting with the title – ‘emerging role’! Most of these studies were published much earlier and reviewed several times too. What are ‘emerging’ in this case? Are there any new developments in the knowledge? If yes, they should be specifically highlighted, and the sections of the manuscript should be reframed accordingly.
3. Although the authors talk about zinc and neuroscience, they failed to draw basic informative layouts. For example – what is the concentration of zinc in different brain regions; does they differ among various cell types; does they change with normal aging; what is the source of zinc in brain; zinc metabolism in brain; etc etc. Basic information should be put to make these points clear, then evaluate how different neuropathological situations alter the zinc machinery in different ways!
4. Several sections are misleading, especially accompanied with wrong reference. For example – line39 - It is possible for dietary zinc supplementation to prevent certain diseases associated with….. – how is this possible since zinc cannot cross the BBB! (PMID: 35740910)
5. Line45: How can the authors justify the statement ‘and it may serve as a cut-in point for investigating the pathophysiology of many neurological diseases.’
6. Line 61 – ‘More than 95% of zinc is present in fat-free 61 tissues such as the testes, muscle, liver, and brain [13]’ – This is wrong since brain and testes are organs which contain very high fat content.
7. Line 68: what is the normal level of zinc?
8. Line71: The reference number 19 is a study on E. coli! These cannot be substantiated as a study to understand neurological aspects!
9. Section3: This section does not contain the information which it is supposed to have – distribution of zinc! The authors did not convey the distribution pattern, regionalities, speciation of zinc, etc.
1. Line 74: ‘The amount of zinc in brain tissue is second only to iron in terms of trace metal concentration [14,22,23]’ – The authors did not consider which species they are talking about. Moreover, different regions of the brain has different levels of trace metals. Zinc is not the second most available trace metal, and generalization of such statements are misleading (PMID: 30891444)
1. Line90-92: ‘the stimulation of AMPA and NMDA receptors accelerated Zn2+ influx into neurons [33,34]’ – misleading statement – since the original studies showed that zinc was bound to NMDA in the cell membrane. The original study has not shown that stimulation of NMDA/AMPA increases zinc influx.
1. Line105/106: ‘Prenatal and postnatal ZnSO4 supplementation enhanced spatial memory and elevated blood zinc levels in rat pups [39].’ – In the original study, the zinc supplementation was done to the mother and not to the pups. The authors failed to convey the appropriate information.
1. There are several more instances which the authors needs to carefully evaluate while preparing their revised manuscript for future submission.
1. Several references were mis-quoted and wrongly cited. For example ref 57 has nothing to do with PARP-1 chelation; ref 54 does not state that stroke is the 2nd ranked cause of death globally; ref 89 does not state that there are 37 million AD patients globally; and many more!
1. Fig 1: RNA is not double stranded! Fig2. Seems like to convey that it is only through ROS that zinc modulates neurotoxicity – which is quite out of place!
1. The discussion section is incomplete and does not put forward any novel discussion.
Author Response
In response to Reviewer 1:
The review by Li et al., ‘The emerging role of zinc in neurological diseases’ has tried to address the role of zinc in various neurological conditions and highlight how zinc can be targeted for therapeutic outcomes. Although the topic is relevant, but there are certain aspects which the authors have to be careful. The quality of the manuscript needs to be worked upon, since several sections are redundant and sometimes contain misleading aspects.
- Novelty aspects: Although a review is always great to read, the authors must take what they are offering in their manuscript! Reviewing the role of zinc in various physiological and neuropathological conditions has been done many times in the past (PMID: 22473811, 22072659, 22457776, 33255662, 7082716, 23882214, and many more) and also some in the present special issue "Zinc in Health and Disease Conditions". The authors did not put forward a solid motivation for writing this review, which in such a scenario becomes redundant!
Response: Indeed, zinc in neurological conditions has been reviewed several times, as the Reviewer rightly points out. Given its importance in neurology, our opinion is that we should keep zinc in the limelight, so that much more can be researched on zinc’s activities and how to overcome its neurotoxic potential. Thus, we have carefully revised the manuscript in light of the Reviewer’s concern, and expanded and added more novel comments and insights (shown in red ink) throughout the manuscript. Our highlight of zinc deficiency or excess being associated with different neurological conditions should help to keep focus on zinc in neurology. We hope the Reviewer agrees with our reasoning.
- Irrespective of what is mentioned above, the present manuscript has many disparities. Starting with the title – ‘emerging role’! Most of these studies were published much earlier and reviewed several times too. What are ‘emerging’ in this case? Are there any new developments in the knowledge? If yes, they should be specifically highlighted, and the sections of the manuscript should be reframed accordingly.
Response: These critiques are fair. As zinc has been discussed in neurology for some time, we have changed ‘emerging’ in the title to ‘important’. We have also added several 2022 references so as to show that this field has continued to add new data to the role of zinc in neurological conditions.
- Although the authors talk about zinc and neuroscience, they failed to draw basic informative layouts. For example – what is the concentration of zinc in different brain regions; does they differ among various cell types; does they change with normal aging; what is the source of zinc in brain; zinc metabolism in brain; etc etc. Basic information should be put to make these points clear, then evaluate how different neuropathological situations alter the zinc machinery in different ways!
Response: Thank you for pointing this out. We have now added several sentences (red-inked) in the sections of ‘Distribution of zinc in the brain’, ‘Role of zinc in promoting redox homeostasis’, and ‘Role of zinc on immunity in the CNS’. Indeed, these additions now introduce the basic information better, and set a better link to the subsequent discussions of zinc in various neurological disorders. The manuscript now flows better, thank you.
- Several sections are misleading, especially accompanied with wrong reference. For example – line39 - It is possible for dietary zinc supplementation to prevent certain diseases associated with….. – how is this possible since zinc cannot cross the BBB! (PMID: 35740910)
Response: Thank you for this appropriate critique. We have corrected where possible, and have tried to avoid misleading sentences. With regards to the previous incorrect statement of dietary zinc preventing diseases, we have now changed that to: ‘Maintaining zinc homeostasis is crucial for normal brain function, and zinc deficiency or overload may contribute to brain injury and exacerbate neurological conditions [8].’ (line 38-40).
- Line45: How can the authors justify the statement ‘and it may serve as a cut-in point for investigating the pathophysiology of many neurological diseases.’
Response: The Reviewer is indeed correct. We have now softened that sentence to ‘it appears relevant to investigate its potential contributions to many neurological diseases’ (line 45-46). The next sentence then described the intent of the review.
- Line 61 – ‘More than 95% of zinc is present in fat-free 61 tissues such as the testes, muscle, liver, and brain [13]’ – This is wrong since brain and testes are organs which contain very high fat content.
Response: Thank you for the remark. We have corrected this statement in line 61-62, to ‘Zinc is highly concentrated (>95%) in fat-free body masses, mainly in intracellular compartments [19].’
- Line 68: what is the normal level of zinc?
Response: In lines 50-51, we have now added: ‘The adult human brain contains approximately 0.04 g of zinc, accounting for 1.5% of the total body zinc [13,14].’
- Line71: The reference number 19 is a study on E. coli! These cannot be substantiated as a study to understand neurological aspects!
Response: Thank you for pointing this out. We have now corrected this on line 71.
- Section3: This section does not contain the information which it is supposed to have distribution of zinc! The authors did not convey the distribution pattern, regionalities, speciation of zinc, etc.
Response: We apologize for this omission. Accordingly, we have expanded Section 3 (’Distribution of zinc in the brain’) to include this new information. Where appropriate, other sections (e.g. section 3.1, on recommended daily allowance during pregnancy) also contain such new addition.
- Line 74: ‘The amount of zinc in brain tissue is second only to iron in terms of trace metal concentration [14,22,23]’ – The authors did not consider which species they are talking about. Moreover, different regions of the brain has different levels of trace metals. Zinc is not the second most available trace metal, and generalization of such statements are misleading (PMID: 30891444)
Response: Thank you for pointing this out. We have modified the misleading statements and substantially expanded the description of the distribution of zinc in the brain (paragraphs 1 and 2 of Section 3).
- Line90-92: ‘the stimulation of AMPA and NMDA receptors accelerated Zn2+ influx into neurons [33,34]’ – misleading statement – since the original studies showed that zinc was bound to NMDA in the cell membrane. The original study has not shown that stimulation of NMDA/AMPA increases zinc influx.
Response: Thank you for pointing this out. We have removed the misleading statement. We now state: ‘Zinc enters neurons through three main pathways: α-amino-3-hydroxyl-5-methyl-4-isoxazolepropionate-acid /kainite (Ca2+-A/K) AMPA/KA channels [39], voltage-dependent channels, and N-methyl-D-aspartate (NMDA) receptor-gated channels [40].’ (line 96-98)
- Line105/106: ‘Prenatal and postnatal ZnSO4 supplementation enhanced spatial memory and elevated blood zinc levels in rat pups [39].’ – In the original study, the zinc supplementation was done to the mother and not to the pups. The authors failed to convey the appropriate information.
Response: Thank you for this scrutiny. We have now corrected this on lines 113-115 to ‘Wistar females during pregnancy and lactation received a drinking water solution of ZnSO4 at an estimated dose of 16 mg/kg. Behavioral tests showed enhanced spatial memory and increased blood zinc levels in the pups [48].’
- There are several more instances which the authors needs to carefully evaluate while preparing their revised manuscript for future submission.
Response: Thank you for pointing this out. We have now extensively revised the manuscript and hope that the current form is improved much more acceptable for publication. We appreciate the Reviewer’s time and attention to the details.
- Several references were mis-quoted and wrongly cited. For example ref 57 has nothing to do with PARP-1 chelation; ref 54 does not state that stroke is the 2nd ranked cause of death globally; ref 89 does not state that there are 37 million AD patients globally; and many more!
Response: We apologize for this and have made the necessary corrections in the revised version.
- Fig 1: RNA is not double stranded! Fig2. Seems like to convey that it is only through ROS that zinc modulates neurotoxicity – which is quite out of place!
Response: Thank you for pointing this out. Figure 1 is now corrected. In the legend to figure 2, we now clarify that while we show a central role for ROS, other mechanisms of zinc toxicity are described in the text. This hopefully will avoid the perception that ROS is the only mechanism of zinc toxicity.
- The discussion section is incomplete and does not put forward any novel discussion.
Response: Thank you for this fair critique. We have added several sentences to the final ‘Summary and Conclusions’ section that hopefully add perspectives to the field. Overall, we are grateful for the thoroughness of the Reviewer. We believe that the revised manuscript is now much improved as a result.
Reviewer 2 Report
Article Review: The emerging role of zinc in neurological diseases
Summary
Sections 1 and 2 provided important information for the reader to establish the roles of zinc and the homeostasis of zinc within cells and tissues throughout the body.
The authors then continued to discuss the tissue distribution of zinc, particularly in the central nervous system.
Overall the authors provide fine arguments and sufficient citations. See recommendations below.
Recommendations:
Rename Section 3.1 Role of zinc in neurogenesis.
Is there a role for zinc in neuronal migration?
Section 3.2 Combine paragraphs 1 and 2.
Explain point c for controlling redox signaling directly or indirectly. Through protein bound functions (super oxide dismutases?)
3.3 Role of zin on immunity in the CNS
Reference 53 is a review article. Where are the primary sources to support this statement?
The grammar in the first paragraph page 4, last sentence needs to be addressed.
Section 4.1
There needs to be a separation of paragraphs lines 158 and 159 to separate positive and negative effects of zinc.
Understanding the role of zinc supplementation on stabilizing endothelial integrity is the message of the first paragraph. Alternatively, the second paragraph discuss the deleterious effect of free zinc after neuronal injury due to an ischemic stroke. This concept then continues into the next paragraph.
Reference 71 seems inconsistent and is not explained for the role of zinc in the mitochondria.
What is being written about with reference 72? Inside an early work?
Section 5.2 The first sentence of the third paragraph on page 6, line 265 is confusing.
Are the authors intending to propose the necessity for balance of zinc homeostasis? Too much free zinc is deleterious because of toxicity whereas insufficient zinc also results in neuronal death due to apoptosis?
Section 5.4 Depression
Which antidepressant is being referenced in paragraph two of this section? Line 318.
Author Response
In response to Reviewer 2:
Summary
Sections 1 and 2 provided important information for the reader to establish the roles of zinc and the homeostasis of zinc within cells and tissues throughout the body.
The authors then continued to discuss the tissue distribution of zinc, particularly in the central nervous system.
Overall the authors provide fine arguments and sufficient citations. See recommendations below.
Response: Thank you for the kind comments.
Recommendations:
Rename Section 3.1 Role of zinc in neurogenesis.
Response: Thank you for the suggestion and we have done so in line 100.
Is there a role for zinc in neuronal migration?
Response: Yes, Zinc plays a role in neuronal migration, and we have added this in line 101-103 to ‘Zinc plays critical roles in neuronal proliferation and differentiation, neuronal migration and axonal growth during neurodevelopment [30].’
Section 3.2 Combine paragraphs 1 and 2.
Response: Thank you for this excellent suggestion and we have done so.
Explain point c for controlling redox signaling directly or indirectly. Through protein bound functions (super oxide dismutases?)
Response: Thank you for pointing this out. We have revised and explained how the control of redox signaling occurs to ‘and controlling redox signaling in by altering enzyme activity, binding interactions, and molecular chaperone activity [58].’ (line 131-132)
3.3 Role of zin on immunity in the CNS
Reference 53 is a review article. Where are the primary sources to support this statement?
Response: Thank you for pointing this out. We have added the primary article in line 151.
The grammar in the first paragraph page 4, last sentence needs to be addressed.
Response: Thank you for the suggestion and we now state: ‘PARP-1, which can chelate zinc, have been shown to minimize neuronal death in rats with ischemic stroke [70] and other neurodegenerative diseases [71].’ (line 157-158)
Section 4.1
There needs to be a separation of paragraphs lines 158 and 159 to separate positive and negative effects of zinc.
Response: Thanks for careful reading and we have done so.
Understanding the role of zinc supplementation on stabilizing endothelial integrity is the message of the first paragraph. Alternatively, the second paragraph discuss the deleterious effect of free zinc after neuronal injury due to an ischemic stroke. This concept then continues into the next paragraph.
Reference 71 seems inconsistent and is not explained for the role of zinc in the mitochondria.
Response: Thank you for your suggestion. We have modified this section to make it more appropriate in line 191-192 to ‘In an experimental model of ischemic stroke, normobaric hyperoxia treatment reduces zinc accumulation in penumbral tissues, thus reducing ischemic injury [84].’
What is being written about with reference 72? Inside an early work?
Response: Thank you for the remark. We have corrected this statement to ‘Serum zinc concentrations were considerably lower in ischemic stroke patients compared to age- and sex-matched healthy controls [85].’ (line 194-196)
Section 5.2 The first sentence of the third paragraph on page 6, line 265 is confusing.
Are the authors intending to propose the necessity for balance of zinc homeostasis? Too much free zinc is deleterious because of toxicity whereas insufficient zinc also results in neuronal death due to apoptosis?
Response: Thank you for your suggestion. The first sentence now reads: ‘Under pathological conditions, neurotoxic levels of free zinc can accumulate in neurons and lead to neuronal damage’ in line 293-294. We have also modified this section to make it more understandable. Indeed, too much free zinc is deleterious because of toxicity whereas insufficient zinc also results in neuronal death.
Section 5.4 Depression
Which antidepressant is being referenced in paragraph two of this section? Line 318.
Response: Thank you for pointing this out. We have now added the antidepressant:
Escitalopram and venlafaxine in line 354-355.
Reviewer 3 Report
The review article Biomolecules-2027454 entitled “The emerging role of zinc in neurological diseases” is interesting. However, some very important issues were presented in this work too generally and they should be addressed.
All questions that should be addressed are mentioned below.
Point-by-point remarks to the Authors
1) In the last sentence of the Abstract section the Authors have stated that in this review article they explored the mechanisms of brain zinc balance, the role of zinc in neurological diseases, and the new effective strategies for the prevention and treatment of these diseases. The issue of the role of zinc in the new effective strategies for the prevention and treatment of neurological diseases was presented too generally. More detailed data from studies in humans should be provided.
2) Line 60: “bodily fluids” whether “body fluids”
3) Lines 61-62: This sentence needs correction because the brain is not a fat-free tissue as it was stated.
4) Line 65: “an essential trace element” – redundant statement, repetition (please see line 33)
5) Line 79: the term “free zinc” is unfortunate in a scientific article.
6) Line 87: “for transport and metabolism” – transport is one of the metabolic processes in the body
7) Line 83: a calcium- and impulse-dependent way [29].
8) Line 89: “Ca2+-permeable” not “Ca2+-permeable”
9) I would like to suggest the Authors change the title of subsection 3.1 to “Role of zinc in neurogenesis”
10) Line 97: “of proteins, enzymes, hormones, and growth factors [35].” – enzymes are proteins
11) Line 106: ZnSO4 – all chemical formulas and abbreviations should be explained at the place of their first use in the text. The same refers to line 114 “reactive oxygen species (ROS), such as O2−, H2O2, and OH-“. Moreover, I think the Author meant superoxide radical (O2·-) (not O2−) and hydroxyl radical (·OH) but not hydroxyl group (OH-). Please correct in Figure 2.
12) Lines 107-110: “Zinc deficiency during pregnancy and breast-feeding can impede the growth of newborns, increasing the risk of dwarfism, impaired learning memory function, and delayed mental development in offspring, whereas zinc supplementation can improve premature newborns develop [40,41].” More detailed data are necessary, how much zinc deficiency can impede the growth of newborns, and how the level of zinc supplementation (and in what form?) can improve premature newborn’s development?
13) Line 130: „is a trace element” - repetition
14) Line 130-132: At what zinc deficiency patients show greater sensitivity to numerous diseases? The statement that zinc deficiency increases the sensitivity of patients to numerous diseases is too general. Moreover, it is interesting to how kind of diseases.
15) Line 182: “the researchers postulated that low blood zinc levels were associated”, generally zinc is determined in the serum. Moreover, what the Authors meant by writing “low blood zinc levels”. Such information is very imprecise and will be unclear for a reader. The word “concentration” is more appropriate than “level”.
16) Line 183-185: “In a case-cohort study, blood zinc concentrations were inversely linked with the incidence of ischemic stroke, especially in women, indicating that low zinc levels may be a risk factor for ischemic stroke [73].” How zinc concentrations are related to the increased risk of ischemic stroke and how these concentrations refer to that noted in the general population?
17) Line 201: “Patients with ICH had lower zinc levels than normal controls [86].” How higher?
18) Lines 205-206: “The findings indicated that zinc deficiency is associated with a more severe course of the disease.” How was the deficiency of zinc?
19) Section 5.1: How are the concentrations of zinc in the serum in AD patients? Some data on zinc concentrations in the serum in AD patients are necessary?
20) Lines 235-236: “Excess zinc is also linked to hyperphosphorylated tau aggregation in AD pathogenesis at the molecular level.” What kind of excess?
21) Line 269-270: “Similar to this, dietary zinc deficiency in rodents inhibited the proliferation of neural precursor cells and hippocampus neurogenesis in animal studies [25,114].” How kind of deficiency? How the findings in an animal model may be extrapolated into humans?
22) Lines 274-275: “In a clinical trial, 68 patients with severe brain injury were randomly allocated to the zinc supplementation versus routine zinc therapy group.” In what form and amount zinc was supplemented?
23) Lines 287: “Pretreatment with high dosages of zinc” – too general statement, who were the doses of zinc?
24) Lines 290-291: “Therefore, zinc can reduce the severity of seizures when given at the appropriate dosage [122,123].” What is the appropriate dosage?
25) Line 297: “patients” whether “children with intractable epilepsy”?
26) Line 313: “have decreased serum zinc levels [131].” – how the concentration of zinc was decreased?
27) Line 319-321: “Importantly, zinc supplementation reversed the behavioral changes associated with depression, indicating that zinc deficiency may contribute to drug resistance [136].” How zinc compounds, in which doses, and for how long were supplemented?
28) Lines 322-323: How concentrations of zinc in the serum were noted in patients suffering from depression?
29) Lines 323-325: “A meta-analysis revealed that patients with depression had lower peripheral blood zinc concentrations, and the severity of depression correlates with the relative zinc deficiency [137]. Similar findings were observed in postmenopausal women [138], college-aged women [139], and hemodialysis patients with end-stage renal disease [140]. Additionally, women with poor dietary or supplementary zinc consumption were more likely to suffer depressive symptoms, whereas men had no significant correlation [141].” The text is too general. More detailed data on zinc deficiency/dietary intake/supplementation should be provided.
30) It is necessary to correct the “Summary and conclusions” taking into account the above remarks.
31) English language correction of the article is necessary. For example:
- “In some cases, however, changes in zinc status can have different effects neurological diseases, like stroke, neurodegenerative diseases, traumatic brain injuries, depression, etc.” (lines 22-24)
- “Adult male rats subjected to middle cerebral artery occlusion (MCAO), the injection of zinc chelator ZnEDTA 14 days later resulted in a considerable decrease in infarct volume, along with an inhibition of neuronal damage and an improvement in neurological function [69].” (lines 169-172).
Author Response
In response to Reviewer 3:
The review article Biomolecules-2027454 entitled “The emerging role of zinc in neurological diseases” is interesting. However, some very important issues were presented in this work too generally and they should be addressed.
All questions that should be addressed are mentioned below.
Response: Thank you for the comment that the article is interesting.
Point-by-point remarks to the Authors
- In the last sentence of the Abstract section the Authors have stated that in this review article they explored the mechanisms of brain zinc balance, the role of zinc in neurological diseases, and the new effective strategies for the prevention and treatment of these diseases. The issue of the role of zinc in the new effective strategies for the prevention and treatment of neurological diseases was presented too generally. More detailed data from studies in humans should be provided.
Response: Thank you for pointing this out. We have now added several sentences (noted in red ink) throughout the manuscript on findings of zinc in human studies.
- Line 60: “bodily fluids” whether “body fluids”
Response: Thank you for pointing this out. We have changed to “body fluids” in line 60.
- Lines 61-62: This sentence needs correction because the brain is not a fat-free tissue as it was stated.
Response: Thank you for the remark. We have corrected it to ‘Zinc is highly concentrated (>95%) in fat-free body masses, mainly in intracellular compartments [19].’ in line 61-62.
- Line 65: “an essential trace element” – redundant statement, repetition (please see line 33)
Response: Thank you for pointing this out. We have deleted the redundant statement in line 65.
- Line 79: the term “free zinc” is unfortunate in a scientific article.
Response: Thank you for the remark. We have corrected it to “zinc ions (Zn2+)” in line 86.
- Line 87: “for transport and metabolism” – transport is one of the metabolic processes in the body
Response: Thank you for the remark. We have corrected this statement to “Given that zinc is unable to diffuse across the cell membrane, specialized transporters or membrane channels are required for its entry into cells [38].” in line 93-94.
- Line 83: a calcium- and impulse-dependent way [29].
Response: Thank you for the remark. We have made the correction in line 91.
- Line 89: “Ca2+-permeable” not “Ca2+-permeable”
Response: Thank you for the remark. We have now changed that to “Zinc enters neurons through three main pathways: α-amino-3-hydroxyl-5-methyl-4-isoxazolepropionate-acid /kainite (Ca2+-A/K) AM-PA/KA channels [39], voltage-dependent channels, and N-methyl-D-aspartate (NMDA) receptor-gated channels [40].” in line 96-98.
- I would like to suggest the Authors change the title of subsection 3.1 to “Role of zinc in neurogenesis”
Response: Thank you for the suggestion and we have done so in line 100.
- Line 97: “of proteins, enzymes, hormones, and growth factors [35].” – enzymes are proteins
Response: Thank you for the remark. In line 103-104, we have corrected this statement to “It is necessary for the synthesis of a wide range of proteins, hormones, and growth factors [44].”
- Line 106: ZnSO4 – all chemical formulas and abbreviations should be explained at the place of their first use in the text. The same refers to line 114 “reactive oxygen species (ROS), such as O2−, H2O2, and OH-“. Moreover, I think the Author meant superoxide radical (O2-) (not O2−) and hydroxyl radical (·OH) but not hydroxyl group (OH-). Please correct in Figure 2.
Response: Thanks for careful reading and pointing this out. We have now added “Wistar females during pregnancy and lactation received a drinking water solution of ZnSO4 at an estimated dose of 16 mg/kg.’ in line 113-114. Accordingly, we now state: ‘Reactive oxygen species (ROS) are chemically reactive chemical entities containing oxygen such as O2·-, H2O2, and ·OH, which are constantly produced in vivo under aerobic conditions [52,53].” in line 123-125.
- Lines 107-110: “Zinc deficiency during pregnancy and breast-feeding can impede the growth of newborns, increasing the risk of dwarfism, impaired learning memory function, and delayed mental development in offspring, whereas zinc supplementation can improve premature newborns develop [40,41].” More detailed data are necessary, how much zinc deficiency can impede the growth of newborns, and how the level of zinc supplementation (and in what form?) can improve premature newborn’s development?
Response: Thank you for your suggestion. This section has been expanded in line 117-118 to “The US recommended dietary allowance for zinc for women during pregnancy is 11.5±1.75 mg/d [49]. Zinc deficiency (<56 μg/dL) during pregnancy and breastfeeding can impede the growth of newborns, increasing the risk of dwarfism, impaired learning memory function, and delayed mental development, whereas zinc supplementation can improve the development of premature newborns [50,51].”
- Line 130: „is a trace element” - repetition
Response: Thank you for your suggestion. We have deleted the duplicate expression.
- Line 130-132: At what zinc deficiency patients show greater sensitivity to numerous diseases? The statement that zinc deficiency increases the sensitivity of patients to numerous diseases is too general. Moreover, it is interesting to how kind of diseases.
Response: Thank you for your suggestion. We have modified this section to make it more appropriate in line 145-148 to “Patients with zinc deficiency show increased susceptibility to various pathogens [62]. Regarding the human immune system, the innate and adaptive immune system are both regulated by zinc-finger-bearing transcription factors. Hence, a direct as well as an indirect role of zinc in altering intracellular signaling can be anticipated [63].”
- Line 182: “the researchers postulated that low blood zinc levels were associated”, generally zinc is determined in the serum. Moreover, what the Authors meant by writing “low blood zinc levels”. Such information is very imprecise and will be unclear for a reader. The word “concentration” is more appropriate than “level”.
Response: Thanks for careful reading and pointing this out. In lines 194-196, we have now added: “Serum zinc concentrations were considerably lower in ischemic stroke patients compared to age- and sex-matched healthy controls [85].”. And we have changed “level” to “concentration” in line 198.
- Line 183-185: “In a case-cohort study, blood zinc concentrations were inversely linked with the incidence of ischemic stroke, especially in women, indicating that low zinc levels may be a risk factor for ischemic stroke [73].” How zinc concentrations are related to the increased risk of ischemic stroke and how these concentrations refer to that noted in the general population?
Response: Thank you for pointing this out. We have modified and added the related content in the revised version in line 200-204 to “This might be related to the function of zinc in the metabolism of sex hormones and reproductive cycle but the mechanism underlying the possible interaction is still unclear and warrants further investigation [87]. It appears that ischemic stroke could be a condition that would benefit from preventative zinc supplementation [86].”
- Line 201: “Patients with ICH had lower zinc levels than normal controls [86].” How higher?
Response: Thank you for the remark. We have added up the numerical values “(0.13 ± 0.02 vs 3.17 ± 0.74 μg/dI); P<0.001)” in line 221.
- Lines 205-206: “The findings indicated that zinc deficiency is associated with a more severe course of the disease.” How was the deficiency of zinc?
Response: Thank you for the remark. We have added up the numerical values “(plasma zinc concentration <10 μmol/L)” in line 226.
- Section 5.1: How are the concentrations of zinc in the serum in AD patients? Some data on zinc concentrations in the serum in AD patients are necessary?
Response: Thank you for pointing this out. We have modified and added “There was a twofold increase (25 μg/g) in zinc levels in AD brain tissue as compared with normal age-matched control subjects [108]. Brain zinc accumulation is a prominent feature of advanced AD.” in line 240-243.
- Lines 235-236: “Excess zinc is also linked to hyperphosphorylated tau aggregation in AD pathogenesis at the molecular level.” What kind of excess?
Response: Thank you for the remark. It means “excess chronic zinc supplementation”. We have explained this in next paragraph in line 258-264.
- Line 269-270: “Similar to this, dietary zinc deficiency in rodents inhibited the proliferation of neural precursor cells and hippocampus neurogenesis in animal studies [25,114].” How kind of deficiency? How the findings in an animal model may be extrapolated into humans?
Response: Thank you for pointing this out. We have modified and added the related content in line 299-301. This now says: “Similar to this, dietary zinc deficiency (zinc contents in the zinc-deficient and zinc-adequate control diets were 0.85 ppm and 30 ppm, respectively) in rodents for 5 weeks inhibited the proliferation of neural precursor cells and hippocampus neurogenesis in animal studies [33,130].”
- Lines 274-275: “In a clinical trial, 68 patients with severe brain injury were randomly allocated to the zinc supplementation versus routine zinc therapy group.” In what form and amount zinc was supplemented?
Response: Thank you for the remark. We have added the amount and method of zinc supplementation “(12 mg elemental zinc) versus routine zinc therapy group (2.5 mg elemental zinc), both as the sulfate salt in their total parenteral nutrition solution” in line 306-307.
- Lines 287: “Pretreatment with high dosages of zinc” – too general statement, who were the doses of zinc?
Response: Thank you for the remark. We have modified and added up the numerical values in line 319-321. This now says: “Postnatal day 27 (P27) male Sprague-Dawley rats pretreatment with zinc-deficient diet (2.7 mg/kg) for 4 weeks had long-term adverse effects of seizures.”
- Lines 290-291: “Therefore, zinc can reduce the severity of seizures when given at the appropriate dosage [122,123].” What is the appropriate dosage?
Response: Thank you for your suggestion. We have modified this section in line 330-333 to make it more appropriate. This now says: “High-zinc dietary treatment improved cognitive impairment and reduced regenerative sprouting of hippocampal mossy fibers [138,139].”
- Line 297: “patients” whether “children with intractable epilepsy”?
Response: Thank you for the suggestion and we have done so in line 330.
- Line 313: “have decreased serum zinc levels [131].” – how the concentration of zinc was decreased?
Response: Thank you for pointing this out. We have added “One possible mechanism, involved in the changes in hippocampal functions, behavior and pathophysiology observed in zinc deficiency, may be the activation of the hypo-thalamo-pituitary-adrenocortical (HPA) system [148].” in line 345-348.
- Line 319-321: “Importantly, zinc supplementation reversed the behavioral changes associated with depression, indicating that zinc deficiency may contribute to drug resistance [136].” How zinc compounds, in which doses, and for how long were supplemented?
Response: Thank you for the remark. We have modified and added up the numerical values in line 354-355. This now says: “Importantly, treatment with escitalopram, venlafaxine 10 mg/kg, i.p., or combined escitalopram/venlafaxine (1 mg/kg, i.p.) with zinc (5 mg/kg) for 3 weeks”
- Lines 322-323: How concentrations of zinc in the serum were noted in patients suffering from depression?
Response: Thank you for the remark. We have added up the numerical values “A meta-analysis revealed that zinc concentrations were approximately 1.85 µmol/L lower in depressed patients,” in line 359-360.
- Lines 323-325: “A meta-analysis revealed that patients with depression had lower peripheral blood zinc concentrations, and the severity of depression correlates with the relative zinc deficiency [137]. Similar findings were observed in postmenopausal women [138], college-aged women [139], and hemodialysis patients with end-stage renal disease [140]. Additionally, women with poor dietary or supplementary zinc consumption were more likely to suffer depressive symptoms, whereas men had no significant correlation [141].” The text is too general. More detailed data on zinc deficiency/dietary intake/supplementation should be provided.
Response: Thank you for your suggestion. We have modified this section in line 359-364 to make it more appropriate. This now says: “A meta-analysis revealed that zinc concentrations were approximately 1.85 µmol/L lower in depressed patients, and the severity of depression correlates with the relative zinc deficiency [154]. Furthermore, similar findings were observed in depressed patients such as postmenopausal women (72 ± 14 μg/dL) [155], college-aged women (79.6 ± 30.7 μg/dL) [156], and hemodialysis patients with end-stage renal disease (67.46 ± 29.7 μg/dL) [157].”
- It is necessary to correct the “Summary and conclusions” taking into account the above remarks.
Response: Thanks for careful reading and pointing this out. We have added several sentences to the final ‘Summary and Conclusions’ section that hopefully add perspectives to the field. Overall, we are grateful for the thoroughness of the Reviewer. We believe that the revised manuscript is now much improved as a result.
- English language correction of the article is necessary. For example:
- “In some cases, however, changes in zinc status can have different effects neurological diseases, like stroke, neurodegenerative diseases, traumatic brain injuries, depression, etc.” (lines 22-24)
Response: Thank you for your suggestion. We have rewritten the statement in line 22-24. The sentence now says: “There are several neurological diseases that may be affected by changes in zinc status, and these include stroke, neurodegenerative diseases, traumatic brain injuries, depression.”
- “Adult male rats subjected to middle cerebral artery occlusion (MCAO), the injection of zinc chelator ZnEDTA 14 days later resulted in a considerable decrease in infarct volume, along with an inhibition of neuronal damage and an improvement in neurological function [69].” (lines 169-172).
Response: Thank you for your suggestion. We have rewritten the statement in line 186-189. This now says: “The injection of the zinc chelator ZnEDTA 14 days after middle cerebral artery occlusion (MCAO) in adult male rats led to a significant decrease in infarct volume and neuronal damage, and improvement in neurological function [82].”
Reviewer 4 Report
Reviewer comments and suggestions
The authors in this study summarize the mechanisms of brain zinc balance, the role of zinc in neurological diseases such as stroke, neurodegenerative diseases, traumatic brain injuries, depression, etc. In addition, elaborates the novel effective strategies for the prevention and treatment of neurological disorders.
The paper was nicely written, and a few minor comments needed to consider before publication.
A few concerns/comments needed to be explained/modified.
- Line 36 first time used DNA and RNA so it should be in full form
- Line 38 The authors need to explain here with suitable references
- Line 74-75 please add up the numerical values as well
- Line 81 already defines MMP so here no required full form. Please check these inconsistencies in the MS and line 232 consistent in the manuscript's full name or AD
- Line 125-126 Please explore the paper
- In line 132 the authors need to explain the points only writing is not important.
- Line 174-176 is this line was appropriate in the role of zinc in stroke
- Line 184-185 the authors need to highlight clinical trials as well. Based on my observation the authors need to add a table that may suitable for this paper.
- Line 243 one line was not sufficient to understand this therapy
- Line 324-326 Please add the normal levels of zn in peripheral blood
- Comments for figure 2 The figures needs to be explained well, the authors can get idea from the published paper.
Author Response
In response to Reviewer 4:
The authors in this study summarize the mechanisms of brain zinc balance, the role of zinc in neurological diseases such as stroke, neurodegenerative diseases, traumatic brain injuries, depression, etc. In addition, elaborates the novel effective strategies for the prevention and treatment of neurological disorders.
The paper was nicely written, and a few minor comments needed to consider before publication.
Response: Thank you for these compliments.
A few concerns/comments needed to be explained/modified.
- Line 36 first time used DNA and RNA so it should be in full form
Response: Thank you for your suggestion. We have changed to the full form “deoxyribonucleic acid (DNA) and ribonucleic acid (RNA)”in line 36.
- Line 38 The authors need to explain here with suitable references
Response: Thank you for the remark. We have added references (4-7) on diseases associated with zinc deficiency in line 38, such as chronic kidney disease, cardiovascular disease and depression, etc.
- Line 74-75 please add up the numerical values as well
Response: Thank you for the remark. We have added up the numerical values in line 79-81 to “In this regard, while the iron content in the normal brain is around 0.04 mg/g with a concentration of ~ 720 μM, that of zinc is about 10 μg/g with a concentration of 150 μM [30,31].”
- Line 81 already defines MMP so here no required full form. Please check these inconsistencies in the MS and line 232 consistent in the manuscript's full name or AD
Response: Thanks for careful reading and pointing this out. We have checked the relevant full form in this manuscript.
- Line 125-126 Please explore the paper
Response: Thank you for your suggestion. This section has been expanded in line 134-139. The sentence now says: “The events involved in the regulation of cellular oxidative/antioxidant homeostasis by zinc are diverse and interrelated, including: i) regulation of oxidant production and metal-induced oxidative damage; ii) regulation of glutathione (GSH) metabolism and overall thiol redox status by zinc; and iii) direct or indirect regulation of redox signaling [52]. Thus, the dysregulation of zinc homeostasis may have significant adverse effects.”
- In line 132 the authors need to explain the points only writing is not important.
Response: Thank you for your suggestion. We added an explanation to this part, which now enriches our manuscript, in line 146-148. This now says: “Regarding the human immune system, the innate and adaptive immune system are both regulated by zinc-finger-bearing transcription factors. Hence, a direct as well as an indirect role of zinc in altering intracellular signaling can be anticipated [63].”
- Line 174-176 is this line was appropriate in the role of zinc in stroke.
Response: Thank you for your suggestion. This part confirms that chelated zinc reduces neuronal damage after ischemic stroke. We have rewritten the statement in line 191-192. The sentence now says: “In an experimental model of ischemic stroke, normobaric hyperoxia treatment reduces zinc accumulation in penumbral tissues, thus reducing ischemic injury [84].”
- Line 243 one line was not sufficient to understand this therapy
Response: Thank you for pointing this out. This is the concluding statement; we have now added more of the research data related to zinc and Alzheimer's in paragraph 2, 3 and 4 of section 5.1 (shown in red ink).
- Line 324-326 Please add the normal levels of zn in peripheral blood
Response: Thank you for the remark and we have added “A meta-analysis revealed that zinc concentrations were approximately 1.85 µmol/L lower in depressed patients” in line 359-360.
- Comments for figure 2 The figures needs to be explained well, the authors can get idea from the published paper.
Response: Thank you for your suggestion. We have explained this figure better in the revised version in line 370-376, and in the legend to Figure 2. The legend now says: “The interaction between zinc and ROS in ischemic stroke. In ischemic stroke, zinc is released from sources such as the synaptic vesicles of glutamatergic neurons. Acutely, zinc may elevate ROS through an action on mitochondria. In the chronic phase, zinc accumulation can activate neuronal NADPH oxidases to generate additional oxidative species. The resultant ROS may produce neurotoxicity, further increasing zinc accumulation and ischemic brain injury. Note that while we show a central role for ROS, other mechanisms of zinc toxicity exist, and are described in the text.”
Round 2
Reviewer 1 Report
Thank you for the revised manuscript. Please consider the following points
- Contradicting statements: Line240 states ‘There was a twofold increase (25 μg/g) in zinc levels in AD brain…’ whereas line 395 states ‘Conversely, zinc supplementation may be an efficient therapeutic method for degenerative neurological diseases like Alzheimer's disease, where zinc deficiency appears to lead to diminished long-term memory creation and preservation.’ Which one is a true?
- The discussion still seems to be insufficient. Here are few thoughts that can be included :– (1) situations which may affect zinc levels leading to altered homeostasis – environmental (PMID: 35841924, PMID: 35841924), dietary (PMID: 35841924, https://journals.sagepub.com/doi/epdf/10.1177/156482650102200204), other pathological conditions (if any) ; (2) implication of altered zinc in associated neuronal pathways (PMID: 33597269, 35711243, PMID: 35841924); (3) effective treatment strategies using zinc; (4) since zinc is everywhere in body and it does not cross blood-brain barrier, how can you delivery zinc to brain? (5) If higher zinc is ingested, will it affect other body parts?
- Thoroughly check the manuscript for typos.
Author Response
In response to Reviewer 1:
Thank you for the revised manuscript. Please consider the following points
1. Contradicting statements: Line240 states ‘There was a twofold increase (25 μg/g) in zinc levels in AD brain…’ whereas line 395 states ‘Conversely, zinc supplementation may be an efficient therapeutic method for degenerative neurological diseases like Alzheimer's disease, where zinc deficiency appears to lead to diminished long-term memory creation and preservation.’ Which one is a true?
Response: Thank you for this appropriate critique. Considerable evidence has indicated that there is zinc dyshomeostasis and abnormal cellular zinc mobilization in AD. Most of the studies found that zinc significantly elevated in the brain regions which are the most affected in AD: amygdala, hippocampus and inferior parietal lobule (PMID: 8981312, 9621340, 9667777 and 16832080). Zinc deficiency is also a global problem, especially in the elderly and also in people with Alzheimer's disease. In addition, zinc supplementation reduced the prevalence and symptomatic decline in people with Alzheimer's disease (PMID: 33597269, 28693861, 34727320 and 29914030). Thus, we envisage that zinc may serve twin roles by both initiating amyloid deposition and then being involved in mechanisms attempting to quench oxidative stress and neurotoxicity derived from the amyloid mass. Nevertheless, most experiments are limited to animal models, and it remains debatable whether zinc supplementation is beneficial or harmful for AD patients until more studies clarify this issue. Therefore, we have now corrected this statement in the conclusions.
2. The discussion still seems to be insufficient. Here are few thoughts that can be included:– (1) situations which may affect zinc levels leading to altered homeostasis – environmental (PMID: 35841924, PMID: 35841924), dietary (PMID: 35841924, https://journals.sagepub.com/doi/epdf/10.1177/156482650102200204), other pathological conditions (if any) ; (2) implication of altered zinc in associated neuronal pathways (PMID: 33597269, 35711243, PMID: 35841924); (3) effective treatment strategies using zinc; (4) since zinc is everywhere in body and it does not cross blood-brain barrier, how can you delivery zinc to brain? (5) If higher zinc is ingested, will it affect other body parts?
Response: Thank you for this fair critique. We have added several sentences to the final ‘Summary and Conclusions’ section that hopefully add perspectives to the field. Overall, we are grateful for the thoroughness of the Reviewer. We believe that the revised manuscript is now much improved as a result.
3. Thoroughly check the manuscript for typos.
Response: Thank you for this appropriate critique. We have corrected where possible, and have tried to avoid misleading sentences.
Reviewer 3 Report
The Authors responded to the remarks of the reviewer improving the manuscript. However, there are some questions needed to be addressed.
1) Lines 73-74: What did the Authors mean by writing “At 2–3 g in total, zinc is the second most abundant metal in humans and is distributed 73 unequally throughout different organs and tissues [26].” Why did the Authors state that zinc is the second most abundant metal in the human body? The content of zinc reaches 2-4 g, but zinc is not the second most abundant metal (for example there are 1-1.5 kg of calcium and 24 mg of magnesium).
2) Lines 76-77: The sentence “Zinc is found in 75% to 85% of the red blood cells, 3% to 5% in the white blood cells, and the rest in the plasma [28].” needs correction. The reviewer thinks that the Authors mean that 75-85% of zinc in the blood is present in the red blood cells but they wrote that this element is present in 75-85% of the red blood cells.
3) Line 131: is the “in” necessary?
4) Lines 293-294: The sentence “Under pathological conditions, neurotoxic levels of free zinc can accumulate in neurons and lead to neuronal damage.” needs correction. The statement “neurotoxic levels of free zinc” is inappropriate, because it shows that levels of zinc can be accumulated (zinc, not its levels, may be accumulated). Moreover, what did the Authors mean by writing “free zinc”?
Author Response
In response to Reviewer 3:
The Authors responded to the remarks of the reviewer improving the manuscript. However, there are some questions needed to be addressed.
1. Lines 73-74: What did the Authors mean by writing “At 2–3 g in total, zinc is the second most abundant metal in humans and is distributed 73 unequally throughout different organs and tissues [26].” Why did the Authors state that zinc is the second most abundant metal in the human body? The content of zinc reaches 2-4 g, but zinc is not the second most abundant metal (for example there are 1-1.5 kg of calcium and 24 mg of magnesium).
Response: Thank you for this appropriate critique. We have corrected to “zinc is an abundant transition metal found in high concentrations in mammalian brain” in line 73-74.
2. Lines 76-77: The sentence “Zinc is found in 75% to 85% of the red blood cells, 3% to 5% in the white blood cells, and the rest in the plasma [28].” needs correction. The reviewer thinks that the Authors mean that 75-85% of zinc in the blood is present in the red blood cells but they wrote that this element is present in 75-85% of the red blood cells.
Response: Thank you for this scrutiny. You’re right, and we have revised and explained in line 76-78: “Only a small fraction of zinc circulates in the blood, approximately 80% of which is loosely bound to albumin and 20% is tightly bound to α2-macroglobulin”.
3. Line 131: is the “in” necessary?
Response: Thanks for careful reading and we have deleted the “in”.
4. Lines 293-294: The sentence “Under pathological conditions, neurotoxic levels of free zinc can accumulate in neurons and lead to neuronal damage.” needs correction. The statement “neurotoxic levels of free zinc” is inappropriate, because it shows that levels of zinc can be accumulated (zinc, not its levels, may be accumulated). Moreover, what did the Authors mean by writing “free zinc”?
Response: Thank you for the remark. We have corrected this statement in line 294-296, to “During pathological conditions, excess zinc is rapidly released from presynaptic neuronal vesicles, crosses the postsynaptic membrane via channels or transporters (translocation) and causes neuronal damage and death”. As for “free zinc”, we intend to express that it is a potent killer of neurons and glia with extended exposure to as little as 100 nM leading to neuronal death (PMID: 15891778 and 12547646).